# Evaluating Program Semantics Reasoning with Type Inference in System $F$

**Yifeng He**[1], **Luning Yang**[2], **Christopher Castro Gaw Gonzalo**[1], **Hao Chen**[2]

[1]University of California, Davis, [2]University of Hong Kong

{yfhe,ccgawgonzalo}@ucdavis.edu, l4yang@connect.hku.hk, chenho@hku.hk

## Abstract

Large Language Models (LLMs) are increasingly integrated into the software engineering ecosystem. Their test-time compute (TTC) reasoning capabilities show significant potential for understanding program logic and semantics beyond mere token recognition. However, current benchmarks for code reasoning lack a formal, program-centric deductive framework to ensure sound evaluation, and are incapable of assessing whether models genuinely reason about program semantics or merely exploit superficial associations between natural language and code tokens. To bridge this gap, we introduce TF-Bench, a benchmark designed to evaluate LLM reasoning based on type inference in System $F$, a task we refer to as *program semantics reasoning*. By employing verified transformations to remove semantically irrelevant natural language, we construct TF-Bench$_{pure}$, a purely semantics-driven variant of TF-Bench. Our analysis reveals substantial limitations in state-of-the-art LLMs, with the best-performing LLM (Claude-3.7-sonnet) achieving only $55.85\%$ accuracy on TF-Bench$_{pure}$. Additionally, we propose two novel metrics to assess robustness and the effectiveness of test-time reasoning, underscoring critical limitations in current LLM capabilities and highlighting essential directions for future research.

## 1 Introduction

The ability to understand and write programs has become an important factor in evaluating the intelligence of large language models (LLMs) [1–3]. More recently, test-time compute (TTC), also referred to as *learn to reason*, has become a new scaling paradigm to further improve the performance of generative LLMs via reinforcement post-training [4, 5]. These reasoning models show promising results in software-related tasks. However, popular tasks in code generation and understanding fail to fully reveal LLMs' prowess at reasoning about program semantics, *i.e.* their ability to understand program flows and underlying logic. Hence, popular code generation benchmarks often fail to uncover differences between the training algorithms used to build reasoning LLMs.

Previous work on program reasoning for LLMs often involves generating properties for code to satisfy [6, 7] or predicting the execution behavior of the code [8, 9]. However, these tasks require the LLMs to generate tool-specific properties or behaviors, failing to isolate reasoning capability from knowledge of specific downstream tasks. Furthermore, due to the lack of a clearly defined formal system within the programming language, it is difficult to attribute poor performance to a lack of reasoning capability. Therefore, to rigorously understand the fundamental reasoning capabilities of LLMs as an upstream evaluation, we need a benchmark that: 1. The oracle is well-defined in a formal deductive system, and the result can be produced and verified by the system. 2. The evaluation of program semantics

39th Conference on Neural Information Processing Systems (NeurIPS 2025) Track on Datasets and Benchmarks.

reasoning can be conducted within the programming language itself, isolated from the knowledge requirement about specific downstream tools.

Furthermore, natural language (NL) elements within code, such as comments and the names of identifiers, assist programmers in reading code. However, they do not affect how developers and compilers analyze the logic, data flow, and defects in the code. For instance, renaming functions and variables does not impact how the program operates or the outputs it generates. Previous work [10] showed that the performance of language models on these benchmarks can be heavily influenced by NL elements within code snippets. These NL elements are not related to the program semantics, and can often lead LLM-based approaches to misunderstand programs [11, 12].

Program semantics (*e.g.*, denotational, operational, and axiomatic) of programming languages describe program behavior, as viewed by the compiler. They are invariant to the NL components of programs. By contrast, a programmer's understanding of a program is influenced considerably by its NL components. We call this the *cognitive semantics* of a program. The structural logic behind the programs depends on the program semantics, not the cognitive semantics. The gap between these two types of semantics often leads to bugs and security vulnerabilities in LLM-based software applications. We are not aware of any existing benchmark that addresses this gap in the ability of LLMs to reason about program semantics, a task we refer to as *program semantics reasoning*.

As a first step, we use type inference, or predicting the type signature from function implementations, as a task in reasoning on program semantics. There are three main benefits of using type inference for this evaluation: 1. Type inference is grounded in System $F$ [13], which is a formal natural deduction system, providing a well-defined task to evaluate LLMs' reasoning abilities. 2. Type signatures are unique and can be easily verified. For a provided task, there is only one correct signature, whose correctness can be verified by the Hindley-Milner algorithm [14, 15] implemented in the compiler. 3. Type signatures can be inferred solely from provided dependencies (details are described in Section 3.1). We propose to use type inference as an upstream reasoning task, which is not affected by nor helped by the NL components, unlike downstream applications and generation tasks.

We present TF-Bench, a novel evaluation benchmark for program semantics reasoning. We construct TF-Bench using function-level type inference in Haskell [16] for its strict type system with easy-to-understand syntax. [1] Haskell's core language is based on System $F$ [13], which is parametric polymorphic, ensuring the diversity of tasks in the benchmark. In addition to System $F$, Haskell also supports ad-hoc polymorphism [17] (bounded overloading, System $F_<$ [18]), which is also included in TF-Bench for task diversity. Tasks in TF-Bench are self-contained with function dependencies explicitly provided, so we can remove NL pieces in the benchmark to construct TF-Bench$_{pure}$ without losing the logic chain, making it sound to evaluate reasoning models without the influence of NL-contaminating tokens. Type inference is a form of natural deduction, which makes TF-Bench a naturally suitable benchmark for emerging reasoning LLMs. To the best of our knowledge, TF-Bench is the first work to introduce and address the problem of PL semantics reasoning.

**Our contributions.** **1**. We introduce TF-Bench and its NL-free variant TF-Bench$_{pure}$, a pair of novel benchmarks each containing 188 tasks for program reasoning. [2] **2**. We comprehensively evaluate 64 LLMs with varying parameter sizes (Section 4.3). Our findings indicate that on TF-Bench$_{pure}$, the leading API-access LLM, Claude-3.7-sonnet, only achieves $55.85\%$ accuracy. **3**. Based on the two-variant design of TF-Bench, we propose two novel evaluation metrics: *semantic robustness* (Section 4.4) and *reasoning effectiveness* (Section 4.5). These metrics are useful in understanding the effects of the applied TTC post-training methods in doing deductive reasoning about program semantics. **4**. We provide a detailed analysis of LLMs fine-tuned on code or math corpora (Section 4.6). Our results suggest that LLMs fine-tuned on code tend to overfit NL cues, whereas those fine-tuned on math are more likely to solve the task through reasoning.

---

[1] As similar to mathematical functions with little syntax noise, please see Appendix A.
[2] https://github.com/SecurityLab-UCD/TF-Bench

## 2 Background and Related Work

### 2.1 Learning to understand programs

Large language models (LLMs) have made remarkable advances in the field of natural language processing. To leverage the power of LLMs in software engineering, earlier work has addressed the LLM's ability to learn code representation for a better understanding of programs. Code-related tasks for LLMs can be categorized into generation tasks [19–26] and understanding tasks [27–31]. Predicting programming concepts offers a way to evaluate LLMs' reasoning abilities, utilizing the LLMs to generate program predicates [32], invariants [6], and specifications [33, 34]. While showing potential for downstream tasks, these approaches rely on LLMs' proficiency on using third-party tools and their interfaces, which often results in poor performance. Consequently, evaluations in these settings provide limited insight into LLMs' fundamental reasoning about programming itself. In comparison, TF-Bench offers a language-centric deductive evaluation system, focusing on the fundamental reasoning capabilities of LLMs.

Predicting type annotations using LLMs has been studied to address some limitations of traditional rule-based type inference. This application has been explored through various approaches [35–41]. While a promising downstream application of LLMs, predicting type annotations often relies on memorizing commonly used type names, instead of reasoning about programs in terms of logical structures and semantics. TypeGen [40] made promising progress in this direction by introducing domain-aware chain-of-thought prompts via static analysis, thereby enhancing task realism. However, due to the unsound, gradual, and optional nature of Python's type system [42], their task formulation remains inadequate for rigorously evaluating the reasoning capabilities of LLMs as a benchmark.

### 2.2 Propositions as types

The connection between logical reasoning and programming can be traced back to the Curry-Howard Isomorphism [43, 44], or *propositions as types* [45]. The core idea is that each proposition in logic corresponds to a type in a programming language, and vice versa. For instance, a proposition $A$ implies a proposition $B$ (denoted as $A \implies B$), corresponding to a function mapping from type $A$ to type $B$ (denoted as $A \rightarrow B$). Since implementing a function involves invoking other defined functions to transform inputs of specific types into an output of another (or the same) type, we can interpret these defined functions as logical assumptions. Then the process of implementing the function itself can be viewed as a proof of the proposition. Thus, the concept of *proofs as programs* emerges. This paradigm has been widely applied in security areas like formal verification and machine-aided proving [46–48]. Drawing from type theory, type inference is inherently a reasoning task for LLMs.

### 2.3 Task perturbation to evaluate reasoning robustness

As LLMs demonstrate high performance on various benchmarks, concerns about contamination and overfitting have led researchers to focus on the reasoning processes behind their predictions, rather than the predictions themselves. This line of research often employs task perturbation, where input data is systematically modified to evaluate the model's robustness.

Task perturbation in code understanding involves semantic-preserving code transformations [11, 12, 49]. While this approach has proven effective for supervised tasks with labeled data, it is challenging to adapt to generative models. Furthermore, such methods often lack the flexibility to modify both inputs and desired outputs. TF-Bench overcomes these limitations, as the output type signature can also be rewritten along with its dependencies, thereby requiring the LLMs to have a more comprehensive reasoning about the program.

Recently, similar ideas have been applied to evaluate the reasoning processes of LLMs in mathematics tasks, as studies suggest that LLMs often solve math problems by memorizing patterns from training data rather than engaging in formal reasoning [50]. In this direction, various work [51–53] has been proposed to evaluate the reasoning capabilities of LLMs on math question-answering tasks. Type inference is also an instance of formal natural deduction reasoning; we provide a more detailed discussion on this issue in Appendix E.

# 3 Design of TF-Bench

TF-Bench utilizes *type inference* in System $F$ [13] to test the program reasoning ability of LLMs. The task is to generate the final type signature of a function given its implementation and the type signatures of all invoked functions. We designed a three-stage pipeline to construct TF-Bench, ensuring self-contained type inference tasks, as outlined in Figure 1.

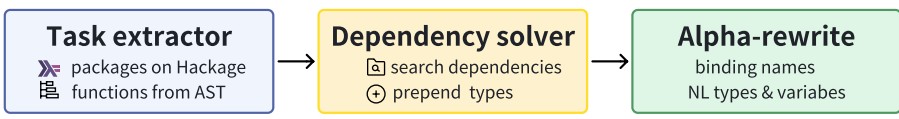

Figure 1: Pipeline to construct TF-Bench. Tasks in TF-Bench are created from Haskell functions with the required type dependencies provided, then rewritten to remove natural language while ensuring soundness.

## 3.1 Benchmark construction

Soundness is crucial when designing benchmarks to evaluate LLMs, particularly in program analysis, logical reasoning, and robustness. In our case, this requires that all type conversions and function mappings in the task are explicitly provided, and that the inference results are decidable within the task's static form. Therefore, the benchmark language must satisfy the following properties: 1. It implements a formal deductive system, specifically the System $F$. 2. It provides concise, standalone type signatures separate from implementation, allowing us to explicitly present type dependencies that form a closed logical chain for reasoning. 3. It should be popular enough so that LLMs have been trained on a sufficient amount of data.

Building a benchmark to understand LLM's program reasoning ability via type inference requires a well-typed language with a soundness guarantee. To this end, we selected Haskell as the foundation for TF-Bench. First, Haskell is based on a variant of the Hindley-Milner polymorphic type system [54], whose soundness has been proven [55]. Second, Haskell is one of the most popular programming languages with type soundness [56, 57], and has been included in popular pre-training datasets such as The Stack V2 [58] with a sufficient amount of training data. Finally, Haskell is a pure functional language where functions are first-class citizens, allowing us to provide a clear and concise task formulation with all type dependencies explicitly provided. Our objective is to evaluate the ability of LLMs to reason about program semantics through type inference, a critical and typical form of reasoning, rather than merely generating code.

We build the tasks in TF-Bench at the function level. We only include the three basic categories of functions in TF-Bench for evaluation purposes: monomorphic, parametric, and ad-hoc functions. Monomorphic functions are the most basic building blocks in programming, where each function is instantiated with a single, fixed type. Parametric polymorphisms are generic functions that can be instantiated with any type. Their type signatures are defined as templates with type variables, enabling flexibility and reusability. Ad-hoc polymorphic functions can be instantiated with any type that satisfies specific constraints. This form of polymorphism is commonly implemented using type classes, subtyping, or overloading, allowing tailored behavior for different types while maintaining type safety. In TF-Bench, we include the definition of the type classes in the task dependencies to ensure that the tasks are *self-contained*. We provide an example for each type of function in Appendix Figure 9.

**Package selection.** We use the standard Haskell Prelude [16] as the foundation for constructing TF-Bench. All functions in the Prelude are defined within the ghc-internal library [59]. The Haskell Prelude, extensively reviewed by the Haskell core library committee, is widely regarded as a reliable and authoritative source for constructing TF-Bench. These functions represent some of the most frequently used code in Haskell and have been adopted in the standard libraries of other programming languages. Therefore, we consider the Prelude a more representative, reliable, and feasible source for building TF-Bench.

**Task extraction.** For each function in the Prelude, we extract its type signature and implementation from the abstract syntax treen (AST) parsed from the source repository. For Ad-hoc polymorphic functions, each typically corresponds to a generic implementation accompanied by multiple specific instantiations. In such cases, we retain the generic implementation as a separate task and include a selection of non-overlapping instantiations in the benchmark. In total, we collected 188 tasks from the Prelude, where 26.6% are monomorphic functions, 32.4% are parametric polymorphisms, and 41.0% are bounded quantification.

**Dependencies solving.** For each candidate task, we conduct dependency solving to provide a self-contained reasoning setup. In type systems, the result of type inference for an implemented function depends on the types of all invoked functions. This notion of type dependencies parallels the concept of assumptions in proof theory and logic. For example, proving the proposition $A \wedge B$ requires the assumption that $A$ is true and $B$ is true. The type dependencies in TF-Bench tasks are assumptions in a proof. To address this, we extract all function invocation nodes from the AST, and retrieve their corresponding type signatures from the Haskell API search engine Hoogle [60]. This step ensures that each task is *self-contained* and establishes a *closed logical chain* for our evaluation. After resolving dependencies, we validate the tasks by compiling them with the oracle, and confirm completeness by manually reviewing all tasks to ensure that no dependencies are missing. We provide an example of a task in TF-Bench in Listing 1.

```
not :: Bool → Bool
(.) :: (b → c) → (a → b) → a → c
span :: (a → Bool) → [a] → ([a], [a])

break p = span (not . p)
-- complete the following type signature
    for `break`
break ::
    (a → Bool) → [a] → ([a],[a])
```

```
f2 :: (t1 → T1) → [t1] → ([t1], [t1])
f3 :: T1 → T1
f4 :: (t1 → t2) → (t3 → t1) → t3→t2

f1 p = f2 (f3 `f4` p)
-- complete the following type signature
    for `f1`
f1 ::
    (t1 → T1) → [t1] → ([t1], [t1])
```

Listing (1) Example task for the `break` function    Listing (2) The task in Listing 1 after alpha-rewrite

Figure 2: Example task in TF-Bench.

## 3.2 Removing natural language from tasks

Previous research indicates that LLMs for code understanding predominantly rely on superficial natural language features rather than genuinely comprehending program semantics [10, 11]. A similar pattern has been observed in generative LLMs applied to mathematical reasoning tasks [53]. Furthermore, the Haskell Prelude, introduced by Jones, predates the knowledge cutoff of most, if not all, existing models. Therefore, related data might already have been included in the training data. To investigate the true ability of generative LLMs in program reasoning, it is essential to identify equivalent forms of the tasks in TF-Bench.

To rigorously assess the LLMs' reasoning ability on program semantics while minimizing the effects of potential data contamination, we designed three rewrite operators aimed at removing natural language elements from TF-Bench. These operators transform code tokens containing natural language components into NL-free equivalents. The rewrite operators are implemented with the type signature `op :: Task → Either String Task` and are verified to be commutative and associative under Kleisli composition [61]. We call the composition of these operators *alpha-rewrite*. Alpha-rewrite does not alter the operational and denotational semantics of the program. The validity of the rewritten tasks is confirmed through successful compilation, ensuring their semantic integrity. We show an example rewritten task of `break` in Listing 2. Our rewrite operators can also be dynamically adjusted with different, and even attacking [11, 62], naming patterns to accommodate further data contamination. We provide an analysis of the impact of each rewrite operator in Appendix D.

**Rewriting NL types.** We refer to all type names in the code that contain natural language elements as NL types. In Haskell, NL types are written with an uppercase letter as their initial character, distinguishing them from type variables to facilitate easier parsing. Examples of

NL types include primitive data types such as `Int`, `Bool`, and `Char`, as well as the names of type classes like `Eq` and `Ord`. We rewrite all NL types using a standardized format: a capital letter `T` followed by a numerical identifier based on their order of appearance in the code.

**Rewriting type variables.** Type variables in generic functions are lowercase letters. By community convention, these variables typically begin with the lowercase letter `a` and proceed alphabetically. We rewrite all type variables using a lowercase letter `t` followed by a numerical identifier corresponding to their order of appearance in the code. Since type variables do not inherently contain any natural language information, applying this rewrite operation to TF-Bench individually should have no impact on the model's performance, even if the model relies on memorized NL elements to answer the questions.

**Rewriting binding names.** Haskell, as a functional programming language, treats everything as a function, including operators and variables. To unify terminology, we use the term *binding* to refer to all of these entities. We standardize all binding names by rewriting them as a lowercase letter `f` followed by a numerical identifier based on their order of appearance. In Haskell, infix and prefix operators are interchangeable. For instance, the prefix notation `add x y` is equivalent to the infix notation `x ` `add` ` y`. Similarly, the infix notation `x + y` is equivalent to the prefix notation `(+) x y`, with the parentheses indicating the operator. To preserve the semantic structure, we maintain the original position of operators during rewriting, ensuring that the transformed tasks adhere to valid Haskell syntax.

## 3.3 Model input

The input prompt is divided into three components: the system prompt, the instruction prompt, and the task prompt. We use the same system and instruction prompts for all models, as depicted in Appendix Figure 5, to guide the models toward a successful generation. We concatenate the system prompt and instruction prompt, and send the concatenation using the 'system' role. However, due to the API difference, such a role is unavailable for OpenAI reasoning models, so we concatenate all three components as a single input.

The task prompt is similar to the examples in Figure 2. As outlined in Section 3.1, we extract functions from Haskell packages to construct the tasks, and provide addition type dependencies for each task. For each extracted function, we concatenate its dependencies separated by new line symbols, and prepend the concatenation to the function definition. Following established work [22, 63, 64], we add an instructive comment in the task to instruct the models to predict the type of the implemented function. Finally, we append a task-specific hook in the form of `` `function_name' :: `` to the end of the input prompt, providing a clear starting point for the LLMs.

## 3.4 Evaluation methodology

**Criteria.** We evaluate whether the generated results match the ground truth by checking for $\alpha$-*equivalence* [65]. By definition, two types are considered $\alpha$-equivalence if their only difference lies in the renaming of bound variables, making them indistinguishable for all practical purposes. Under $\alpha$-equivalence, two polymorphic types are considered equivalent if they are structurally identical, differing only in the naming of type variables. For example, the types `map :: (a → b) → [a] → [b]` and `map :: (c → d) → [c] → [d]` are alpha-equivalent because their type variables are bound in the same order.

Our evaluation pipeline is summarized in Figure 3. First, we design a static analyzer to locate and define all missing types in the ground-truth type signature after alpha-rewrite. With these definitions, along with the ground-truth and LLM-generated type signatures, we construct a proof template. We then formally verify whether the two type signatures are $\alpha$-equivalence. If the proof fails, we treat the LLM's answer as incorrect and include it in our error analysis (see Appendix G for details).

Previous work on using generative models for type inference measures matching up to parametric [40]. This metric considers two types to match if they share the same outermost structure. For instance, `[Int]` and `[Char]` are considered match up to parametric because they share the common outermost type constructor `[]`. However, this evaluation approach

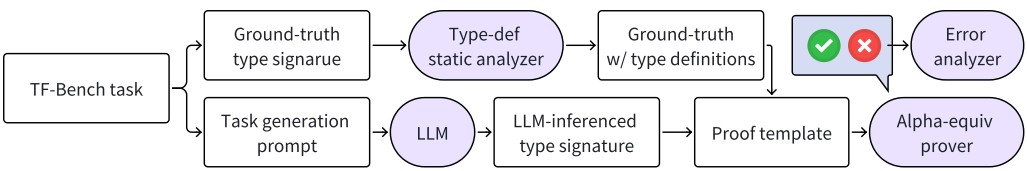

Figure 3: Pipeline to evaluate LLMs on TF-Bench.

is not valid under System $F_<$, since the inner type may not implement the same required trait (or overload the same required operators/functions) [17]. Therefore, we adopt the more rigorous $\alpha$-equivalence metric to evaluate the results of the models in TF-Bench.

**Defining new types.**     After the model generates a type signature, we provide definitions for all referenced types to ensure the proof is self-contained. To this end, we parse the type signatures into ASTs using tree-sitter [66], and propose a static analyzer that reads and classifies all types in the AST. Our analyzer traverses the AST in a depth-first manner, extracting all type classes, type names, and type constructors. For each type constructor, we also record its arity (i.e., the number of type arguments). We then add semantically valid empty definitions for all newly introduced types in the proof. The design of the static analyzer is detailed in Appendix F.1.

**Proving alpha-equivalence.**     Rather than text-level matching, we formally verify that the generated type signature is $\alpha$-equivalence to the ground truth. LLMs need not reproduce the exact type signature, as long as provided the type variables occur in the same order. We prove for equivalence using type equality coercions [67]. To handle $\alpha$-equivalence for polymorphic types with bounded quantification, we enable impredicative types [68, 69] in our proofs. The proof template is given in Appendix F.2. If the proof compiles, the two type signatures are guaranteed $\alpha$-equivalent. This formal approach eliminates false positives and false negatives during evaluation on TF-Bench, thereby ensuring the soundness of our benchmark design. Therefore, we report only accuracy in our experiments.

## 4 Experiments

### 4.1 Experimental setup

Our objective is to evaluate the performance of both state-of-the-art API-access models and open-access models across different sizes. We show the results of top-performing models in Section 4.3, which include the latest models from OpenAI GPT, Anthropic Claude, Google Gemini, DeepSeek, and Qwen. GPT, Claude, and Gemini are commercial API-access models, while DeepSeek and Qwen are open-access models with pre-trained weights available online. However, due to compute limitations, we also access DeepSeek through the API. We also provide a comprehensive list of models and results in the Appendix Table 6. In total, we evaluate 64 models on TF-Bench and TF-Bench$_{\text{pure}}$.

We follow previous studies [2, 70] on code-related tasks to evaluate all models in a zero-shot setting. We use the default temperature and sampling hyperparameters for all models to allow maximum performance. To ensure the validity of the results, we run the models three times and report the average performance, and we observe that all models have standard error $< 0.05$. We utilize Ollama [71] to set up the environment to run the open-access models. We apply straightforward post-processing strategies (Appendix C) to the generation results to ensure consistency and comparability across models.

### 4.2 Research questions

In our experiments, we address the following research questions to understand the performance of various models on TF-Bench and TF-Bench$_{\text{pure}}$: **RQ1** Performance evaluation: We assess the performance of the state-of-the-art API-based and open-source (OSS) models on TF-Bench and TF-Bench$_{\text{pure}}$. **RQ2** Semantics robustness: We examine the robustness of

Table 1: Main evaluation results. TTC: enabled test-time compute reasoning. Acc, $\text{Acc}_{\text{pure}}$: accuracy on TF-Bench and $\text{TF-Bench}_{\text{pure}}$. RS: robustness score. The highest accuracy is in bold, and the second is underlined.

| Model | Version | TTC | Acc | $\text{Acc}_{\text{pure}}$ | RS |
|---|---|---|---|---|---|
| Claude-3.5-haiku | 2024-10-22 | ✗ | 80.85 | 33.51 | 41.45 |
| Claude-3.5-sonnet | 2024-06-20 | ✗ | 85.46 | 48.97 | 57.3 |
| Claude-3.7-sonnet | 2025-02-19 | ✓ | 90.42 | **55.85** | 61.77 |
| GPT-4o | 2024-11-20 | ✗ | 84.57 | 38.12 | 45.08 |
| GPT-O1 | 2024-12-17 | ✓ | 88.30 | 50.00 | 56.63 |
| GPT-O3-mini | 2025-01-31 | ✓ | **90.43** | 48.40 | 53.52 |
| GPT-O3 | 2025-04-16 | ✓ | 81.91 | 52.66 | **64.29** |
| GPT-O4-mini | 2025-04-16 | ✓ | 87.77 | 48.94 | 55.76 |
| DeepSeek-V3 | 2025-03-25 | ✗ | 83.51 | 43.62 | 52.23 |
| DeepSeek-R1 | 2025-01-20 | ✓ | 86.70 | 44.15 | 50.92 |
| Gemini-2.5-flash | Preview-04-17 | ✓ | 83.51 | 51.06 | 61.14 |
| Gemini-2.5-pro | Preview-03-25 | ✓ | 86.70 | 51.60 | 59.52 |
| Qwen3 | 30B-A3B | ✓ | 81.38 | 40.43 | 49.68 |
| | 32B | ✓ | 87.94 | 43.09 | 49.00 |
| | 235B-A22B-FP8 | ✓ | 85.11 | 44.15 | 51.87 |

these models against alpha-rewrites to understand their capability to handle underlying semantics and logical reasoning in programs. **RQ3** Reasoning effectiveness: We investigate the effectiveness of different reasoning strategies by comparing the performance increase after turning on TTC reasoning. **RQ4** Impact of post-training/fine-tuning: We evaluate the performance gains achieved by comparing base models with their fine-tuned versions.

### 4.3 Which are the best performers on TF-Bench?

In our evaluation, the top two models are OpenAI's O3 and Anthropic's Claude-3.7-sonnet, both of which are reasoning models. On TF-Bench, O3-mini and Claude-3.7-sonnet are nearly tied, with accuracy 90.43% and 90.42%, respectively. Among pre-trained open-source models, Qwen3-32B achieves the best performance, correctly solving 87.94% of the tasks.

On $\text{TF-Bench}_{\text{pure}}$, with natural language removed, we observe a significant drop in performance across all models. Requiring the models to reason about the code purely based on the program semantics, $\text{TF-Bench}_{\text{pure}}$ serves as a more challenging variant of TF-Bench and offers distinct insights. Claude-3.7-sonnet is the best-performing model on $\text{TF-Bench}_{\text{pure}}$, achieving 55.85% accuracy. O3, the current flagship reasoning model from OpenAI, comes in second on $\text{TF-Bench}_{\text{pure}}$ with 52.66% accuracy. We provide additional evaluation results in Appendix I, and an analysis of the models' error types in Appendix G.

### 4.4 Semantics robustness

Previous studies [10, 11] have demonstrated that code understanding models are vulnerable to adversarial attacks through identifier name perturbations. Our benchmark design, which includes both a baseline version TF-Bench and a pure variation $\text{TF-Bench}_{\text{pure}}$, enables us to systematically measure LLMs' robustness in reasoning about program semantics. In this section, we assess the robustness of LLMs on program reasoning by introducing task perturbations through alpha-rewrites. We define the Robustness Score (RS) as the model's sensitivity to these alpha-rewrites. Let Acc and $\text{Acc}_{\text{pure}}$ represent the performance of a given model $m$ on TF-Bench and $\text{TF-Bench}_{\text{pure}}$, respectively, $\text{RS}(m) = {\text{Acc}_{\text{pure}}(m)}/{\text{Acc}(m)}$.

We present the robustness scores of the top-performing models in Table 1. Among the evaluated models, O3 and Claude-3.7-sonnet achieve the highest robustness scores of 64.29 and 61.77, respectively, mirroring their superior performance on $\text{TF-Bench}_{\text{pure}}$. These robustness scores quantify each model's ability to maintain consistent performance when confronted with alpha-rewrites, serving as a confidence metric for their semantic reasoning capabilities.

## 4.5 Reasoning effectiveness

Test-time compute reasoning (TTC) is a new scaling paradigm for LLMs, offering a pathway to enable reasoning capabilities in LLMs using natural language. Recent research demonstrates its effectiveness across various benchmarks [4, 5, 72]. However, there remains a critical gap in the systematic evaluation of TTC specifically applied to reasoning about programs. While performance improvements on standard generation benchmarks are evident, it is challenging to disentangle whether these gains stem from the model's enhanced reasoning capabilities or simply from further contamination by natural language cues. This distinction is crucial for understanding the true potential and limitations of developing TTC models.

Our novel two-variant benchmark design, consisting of TF-Bench and TF-Bench$_{\text{pure}}$, enables a precise assessment of TTC's effectiveness on program semantics. Both benchmarks are grounded in System $F$ reasoning of program semantics, providing a program-centric deductive framework for evaluating LLMs. However, improvements in accuracy on TF-Bench alone could potentially be attributed to contamination or overfitting to natural language elements

Table 2: Reasoning effectiveness of top LLMs.

| Model | TTC | Acc | Acc$_{\text{pure}}$ | RE |
|---|---|---|---|---|
| Qwen3-235B-FP8 | ✗ | 80.49 | 35.64 | 1.37 |
| | ✓ | 86.70 | 44.15 | |
| Claude-3.7-sonnet | ✗ | 87.77 | 46.81 | 3.41 |
| | ✓ | 90.42 | 55.85 | |
| Gemini-2.5-flash | ✗ | 78.19 | 30.32 | 3.90 |
| | ✓ | 83.51 | 51.06 | |

rather than genuine reasoning capabilities. To understand TTC's effectiveness, we propose a metric that differentiates the genuine program reasoning improvements from natural language contamination. We utilize the *ratio of accuracy improvements* on TF-Bench$_{\text{pure}}$ compared to TF-Bench as *reasoning effectiveness* (RE) to evaluate the impact of TTC,

$$\text{RE}(m_{\text{ttc}}, m) = \frac{\text{Acc}_{\text{pure}}(m_{\text{ttc}}) - \text{Acc}_{\text{pure}}(m)}{\text{Acc}(m_{\text{ttc}}) - \text{Acc}(m)} = \frac{\Delta_{\text{pure}}}{\Delta}.$$

For our reasoning effectiveness analysis, we focus on models that allow manual control of the TTC mode to ensure evaluation fairness. Currently, only Claude-3.7-sonnet [73], Gemini-2.5-flash [74], and Qwen3-235B-FP8 [75], served using vLLM [76], support this feature. However, our methodology extends beyond these three models. Future research can apply this analysis to any reasoning models with access to the base model before reinforcement learning. Table 2 presents our reasoning effectiveness results. Gemini-2.5-flash achieves the highest reasoning effectiveness of 3.90, with Claude-3.7-Sonnet showing comparable effectiveness at 3.41. A higher RE indicates that the TTC method more effectively improves reasoning capabilities with less reliance on benchmark contamination. RE < 1 suggests the applied TTC method or reinforcement learning failed to develop actual reasoning capabilities, instead promoting overfitting and benchmark contamination.

## 4.6 Impacts of fine-tuning

Data used for supervised fine-tuning can also impact the reasoning ability of LLMs significantly. In this section, we investigate the effects of fine-tuning on code and math datasets, and compare the *performance change* of LLMs after fine-tuning. We present the results in Table 3. Fine-tuning on code corpus does *not* consistently result in performance improvements. Among the evaluated models, DeepSeek-V2 benefits the most from fine-tuning on code. However, Qwen2.5-7B experiences an absolute decrease of - 4.79 on TF-Bench after fine-tuning on code data. Similarly, Mistral-22B and Qwen2.5-72B exhibit decreases of - 8.51 and - 4.26, respectively, on TF-Bench$_{\text{pure}}$. In contrast, fine-tuning on math data yields positive results across both TF-Bench and TF-Bench$_{\text{pure}}$.

Another interesting observation emerges when analyzing the effects of fine-tuning the same model families on different data. For this analysis, we focus on two model architectures: Mistral and Qwen2. Our experiments reveal that fine-tuning these models on code sometimes leads to a decline in performance, while fine-tuning on math consistently results

Table 3: Result comparison of fine-tuning. FT Corpus: the corresponding fine-tuning corpus. $\Delta, \Delta_{\text{pure}}$: absolute increase in accuracy after fine-tuning.

| FT Corpus | Base Model (FT Model) | Size | Acc | FT Acc | $\Delta$ | Acc$_{\text{pure}}$ | FT Acc$_{\text{pure}}$ | $\Delta_{\text{pure}}$ |
|---|---|---|---|---|---|---|---|---|
| Code | Gemma (CodeGemma) | 7B | 48.94 | 53.19 | + 4.25 | 7.45 | 12.23 | + 4.78 |
| | DeepSeek-V2 (-Coder) | 16B | 29.79 | 55.32 | + 25.53 | 7.98 | 15.96 | + 7.98 |
| | | 236B | 38.30 | 80.85 | + 42.55 | 11.17 | 36.70 | + 25.53 |
| | Mistral (Codestral) | 22B | 61.17 | 63.30 | + 2.13 | 19.68 | 11.17 | - 8.51 |
| | Qwen2.5 (-Coder) | 1.5B | 30.32 | 36.70 | + 6.38 | 6.91 | 9.04 | + 2.13 |
| | | 7B | 65.96 | 61.17 | - 4.79 | 21.28 | 21.28 | 0.00 |
| | | 32B | 74.47 | 82.45 | + 7.98 | 36.17 | 31.91 | - 4.26 |
| Math | Mistral (Mathstral) | 7B | 45.21 | 47.34 | + 2.13 | 7.99 | 15.43 | + 7.44 |
| | Qwen2 (-Math) | 7B | 40.43 | 43.09 | + 2.66 | 3.19 | 10.64 | + 7.45 |
| | | 72B | 63.83 | 71.28 | + 7.45 | 21.81 | 33.51 | + 11.7 |

in performance gains. Furthermore, models fine-tuned on code exhibit much smaller improvements on TF-Bench$_{\text{pure}}$ compared to TF-Bench, leading to RE $< 1$, or even RE $< 0$. By contrast, the same models fine-tuned on math demonstrate greater performance improvements on TF-Bench$_{\text{pure}}$. This finding suggests that fine-tuning on math might enhance the models' abstract deductive reasoning capabilities, which also translates effectively to program reasoning and software-related tasks.

## 5 Conclusion

In this paper, we present TF-Bench, a novel benchmark designed to evaluate the ability of language models for program reasoning. TF-Bench focuses on type inference in System $F$, utilizing Haskell syntax and functions to provide a deductive framework for LLM reasoning evaluation. Focusing on the semantic gap between program semantics and cognitive semantics, TF-Bench includes two novel metrics, providing a more comprehensive evaluation of the robustness of LLM reasoning and the effectiveness of reasoning-focused post-training. TF-Bench aims to inspire further research on evaluating LLMs' reasoning capabilities with respect to the semantics of programming languages.

## Acknowledgment

We thank Yiwen Guo and Caleb Stanford for their valuable feedback on this work. We also thank Boqi Zhao and Hezhi Xie for contributing to the initial experimental setup. This material is based upon work supported by UC Noyce Initiative.

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

# Appendix

## Table of Contents

# A  Motivating example

```
map :: (a → b) → [a] → [b]          map :: (a → b) → [a] → [b]
ord :: Char → Int                   chr :: Int → Char
cs :: [Char]                        xs :: [Int]

xs = map ord cs                     ys = map chr xs
-- complete the following type      -- complete the following type
    signature for `xs`                  signature for `ys`
xs ::  [Int]                        ys ::  [Char]
```

Figure 4: Examples of type inference tasks similar to TF-Bench tasks.

Predicting a function's type signature is an efficient and objective method for demonstrating the understanding of the function's logic. As shown in Figure 4, correctly inferring the types of xs and ys requires deductive reasoning through the logical flow of all functions in the expression. For humans and rule-based type checkers, the first step in solving Figure 4 involves analyzing the higher-order function map. On the left, since the first input to map is the function ord :: Char → Int, we can deduce that the parametric type variable a is instantiated to Char and b is instantiated to Int. This process is often referred to as *type instantiation* or *specialization*. The next step is to verify whether the third input to map matches these deductions. As expected, this parameter is a string literal, which has the type [Char], leading us to conclude that xs should have the type [Int]. The type inference for ys follows a similar reasoning process.

Both examples demonstrate the close relationship between type inference and understanding a function's logic. Testing a language model's ability to infer types not only reflects its understanding of the program but also its proficiency in logical reasoning. As the Curry-Howard Isomorphism suggests, the expression bound to xs serves as proof that xs has the type [Int]. This connection between function implementation and logical reasoning highlights the importance and effectiveness of TF-Bench in providing a sound and fine-grained evaluation of language models.

# B  Prompts

```
Act as a static analysis tool for type inference.
ONLY output the type signature.
Do Not Provide any additional commentaries or explanations.
```

System prompt.

```
Remember that in Haskell:
1. The list type `[a]` is a polymorphic type, defined as `data [] a = [] | (:) a [a]`,
so `(:)` is a constructor for list type.
2. The String type is a list of characters, defined as `type String = [Char]`

# for regular tasks
For polymorphic type variables, you can use type variables like `a`, `b`, `c`, etc.
You should start with `a` and increment the alphabet as needed.

# for pure tasks
For polymorphic type variables, you can use type variables like `t1`, `t2`, `t3`, etc.
You should start with `t1` and increment the number as needed.
```

Instruction prompt

Figure 5: Prompt used in TF-Bench.

## C   Post-processing

First, since the `String` type in Haskell is an alias for `[Char]`, we replace occurrences of `[Char]` with `String` in both the generated responses and the ground truth. Second, some model outputs are enclosed in markdown code blocks, so we remove the top and bottom markdown delimiters for consistency. Third, we observe that when a hook (for example, `xs ::` as in the last line in Appendix Figure 4) is provided in the prompt, some models continue from the hook and generate only the type signature, while others repeat the hook in their response. To standardize the outputs, we remove all instances of the hook from both the ground truth and the model outputs. In general, we avoid complex post-processing and focus on resolving basic formatting issues.

## D   Effects of the rewrite operators

In Section 3.2, we introduce three different rewrite operators on different parts of the task: NL types, type variables, and function names. We hypothesize that since type variables do not contain NL elements, rewriting them should have much less performance impact than NL types and function names. In this section, we analyze their effects on LLMs' performance. To answer our research question, we individually rewrite TF-Bench with the three rewrite operators. We run the three flagship general-purpose models GPT-4-turbo, Claude-3.5-sonnet, and O1-preview.

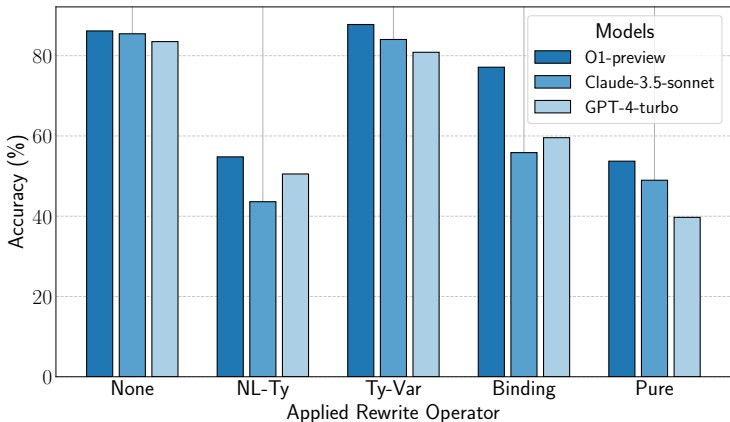

Figure 6: Accuracy on TF-Bench with different rewrite operators. None: the original TF-Bench. NL-Ty: rewriting NL types. Ty-Var: rewriting type variables. Binding: rewriting binding names. Pure: TF-Bench$_{pure}$.

The results presented in Figure 6 indicate that rewriting NL types has the most pronounced effect on model performance, followed by rewriting binding names. In contrast, rewriting type variables has a much smaller impact. This is likely because type variables are inherently generic, and the evaluation considers whether the model's response is alpha-equivalent to the ground truth. Ideally, none of the three rewrite operators should affect the models' performance, which would suggest that the models can effectively reason about programs based on the task's dependencies and the programs' structural relationships.

## E   Bridging type inference to mathematical reasoning

While LLMs have achieved high scores on math question-answering (QA) benchmarks [77, 78], [50] suggests that their apparent success often stems from memorizing patterns in training data rather than reasoning. To address this, task perturbation methods have been developed for math benchmarks to evaluate the underlying reasoning processes.

GSM-Symbolic [52] addresses this issue by creating symbolic templates from the GSM8K dataset [78], where concrete nouns and numbers in the problems are replaced with variables,

requiring LLMs to respond with solutions expressed as combinations of these variables. However, unlike programs, math question-answering tasks are constructed in natural language, making it difficult to systematically determine which tokens can be substituted without altering the problem's semantics. Additionally, this approach may yield different but semantically equivalent answers, complicating the evaluation of the model's correctness and limiting its applicability to other benchmark datasets. These challenges highlight the complexities of designing rigorous evaluations for reasoning in tasks that involve inherent ambiguity.

ReasonAgain [53] adopts a similar approach but, instead of using symbolic variable templates, generates Python programs from math problems using an LLM. The method involves providing different inputs to the generated program, running it to obtain corresponding outputs, and then let an LLM generate new tasks based on the program and the new input-output pairs. While this approach offers the potential to extend to other benchmark datasets, its reliability may be compromised due to the multiple steps that depend on the performance of LLMs, introducing potential sources of error at each stage.

However, the reliance on math QA benchmarks on natural language descriptions makes it challenging to systematically determine which tokens can be substituted without altering the problem's semantics, thereby complicating the verification of perturbation soundness. These methods either cannot easily be applied to other benchmarks or rely on LLMs to modify and generate new tasks, introducing additional risks of uncertainty. In contrast, type inference leverages the benefits of verifiability and soundness from programming and applies them to logical reasoning. Through the Curry-Howard Isomorphism [43, 44], also known as propositions-as-types [45], type inference tasks align well with natural deduction [79]. This enables reliable perturbations that preserve semantic integrity and thus provide a robust framework for evaluating LLMs' mathematical reasoning capabilities. Our findings in Section 4.6, where fine-tuning LLMs on math corpus leads to higher performance improvement on TF-Bench$_{\text{pure}}$ than on TF-Bench, also suggest potential opportunities for future research to explore the mathematical reasoning capabilities of LLMs, aligning with the Curry-Howard Isomorphism.

# F    Proving type equivalence in Haskell

## F.1    Static analysis to define missing types

The first step in preparing the proof is to define all newly introduced types in TF-Bench$_{\text{pure}}$. Because all NL types are rewritten in Section 3.2, these types are not yet defined in Haskell. Thus, we must define them before constructing the proof. To automate this process, we designed a static analyzer that extracts all type names from the rewritten type signature. We employ `tree-sitter` [66] to parse the signature into an AST and identify the signature node. In Haskell, a signature node may contain two children: an optional context node and a mandatory function node, corresponding to type-class constraints and the actual type signature, respectively. We process these nodes separately.

**Context node.**    Extracting type-class constraints is straightforward. The AST context node contains a single level of children, each representing one constraint. We iterate over these child nodes to collect the type names and construct a set of unique names. For each child, we generate an empty type class of the form `class` `<Name>` `a`. Since these constraints always apply to a single type variable, it suffices to define each class with a single parameter a.

**Function node.**    The sub-AST of the function node in Haskell is a binary tree due to currying. All the type names are located at the leaves of the binary tree. To extract them, we perform a depth-first search traversal on the tree. If we reach a terminal node that is a type constructor, we trace back to determine how many type arguments it takes. We define new type constructors using `data` `<Name>` `<vars ... >`, where `<vars ... >` are the type parameters determined by the arity (number of type arguments) of the constructor . Otherwise, we define it as a simple empty type using `data` `<Name>` = `<Name>`. We present a graphical illustration of the AST traversal in Figure 7 and the analysis algorithm in Algorithm 1.

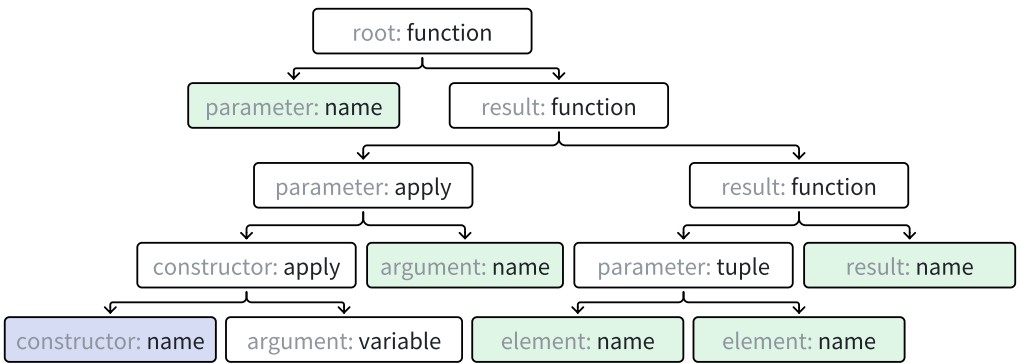

Figure 7: An example AST of f :: `Int` → `Either` a `Char` → (`Int`, `Char`) → `Float`.

Figure 7 illustrates an example abstract syntax tree (AST) for the function signature f :: `Int` → `Either` a `Char` → (`Int`, `Char`) → `Float`. Our static analyzer performs a depth-first traversal from the root node to locate leaf nodes that correspond to type constructors. As outlined in Algorithm 1, when the analyzer encounters a terminal node with type name and field identifier `constructor`, it traces back along consecutive apply nodes to determine the constructor's arity. For instance, in Figure 7, the type constructor `Either` takes two arguments, a and `Char`. If `Either` is rewritten as T1, we define it for the prover as `data` T1 t1 t2. By contrast, if the terminal node is not a type constructor, we define it as a simple empty type using `data` <Name> = <Name>.

---

**Algorithm 1** Static analysis to collect type constructors with applied arity

---

1: **global** Constructors ← dict(name → multi-set of arities)               ▷ Key-value map
2: **function** TOPAPPLY($n$)
3:      $p \leftarrow n$.parent
4:      **return** $\neg(p \neq$ nil $\wedge p$.type = "apply" $\wedge$ CHILD($p$, "constructor") = $n$)
5: **function** PEELAPPLYCHAIN($node$)
6:      $arity \leftarrow 0;\quad cur \leftarrow node$
7:      **while** $cur$.type = "apply" **do**
8:          $arity \leftarrow arity + 1$
9:          $next \leftarrow$ CHILD($cur$, "constructor")
10:          **if** $next =$ nil **then**
11:              **break**
12:          $cur \leftarrow next$
13:      **return** ($cur, arity$)                               ▷ $cur$ is the constructor node
14: **function** VISIT($n$)
15:      **if** $n$.type = "apply" $\wedge$ TOPAPPLY($n$) **then**
16:          ($ctor, arity$) ← PEELAPPLYCHAIN($n$)
17:          Constructors[SRC($ctor$)].ADD($arity$)                  ▷ Src gets the source code
18:      **for all** $c \in n$.named_children **do**
19:          VISIT($c$)

---

## F.2   Constructing proofs of type equivalence

```
{-# LANGUAGE TypeOperators #-}
{-# LANGUAGE ImpredicativeTypes #-}
module Check where

import Data.Type.Equality

-- Some predefined types synonyms to avoid name clashes
type Int_ = Int
type Bool_ = Bool
```

```
type Char_   = Char
type Float_  = Float
type Double_ = Double
data Natural = Natural

$new_types

type TRUTH $truth_vars = $truth_signature
type ANSWER $answer_vars = $answer_signature

proof :: TRUTH $truth_vars :~: ANSWER $truth_vars
proof = Refl
```

Listing 3: Proof for type equivalence

In Listing 3, `Data.Type.Equality` supplies GHC's propositional type equality (`:~:`) with its sole constructor `Refl`, so that this proof type-checks iff TRUTH and ANSWER are definitionally equivalent, *i.e.*, type equality coercions [67]. We use type operators to define symbolic type constructors TRUTH and ANSWER for the ground truth and the model-generated type signature, respectively. We use impredicative types [68, 69] to prove for polymorphisms with bounded quantification. Parts in Listing 3 starting with $ are placeholders that are filled in by our static analyzer described in Section F.1.

## G   Error analysis

Table 4: Error categories for type inference tasks and their definitions.

| Error category | Definition |
| --- | --- |
| OverGeneralization | Chose a type that is too general—used broader polymorphism (e.g., independent input/output type variables) where the most general correct signature requires them to coincide. |
| UnderGeneralization | Added an unnecessary or stronger type-class constraint not justified by the implementation, making the signature more specific than the truly general one. |
| ArgOrderMismatch | Selected the right type variables but arranged them in the wrong parameter order; the arguments are permuted relative to the implementation. |
| ArityMismatch | Provided a type with an incorrect number of arguments, supplying too many or too few relative to the implementation. |
| ConstraintError | Attached incorrect type-class constraints that do not match the implementation's requirements; the wrong constraints were applied to the variables. |
| SyntaxError | Produced an answer that is not a valid Haskell type signature. |
| InstructionFollowing | Failed to follow the instructions given in the prompt. |
| ResponseError | Supplied no answer or an answer entirely unrelated to the task. |

We implemented an LLM-based analyzer to characterize the types of errors produced by the models. We first conducted a manual analysis of the outputs and reasoning summaries from all results generated by Claude-opus-4-1, and grouped its errors into eight categories. These categories and their definitions are summarized in Table 4. The first five categories concern reasoning about program semantics. The sixth category, SyntaxError, captures outputs that are Haskell code but syntactically invalid. The final two categories, InstructionFollowing and ResponseError, concern the model's ability to follow instructions and produce relevant responses. InstructionFollowing denotes cases where the model fails to follow the prompt,

which asks for only the type signature with no additional text. ResponseError covers timeouts or failures to return any answer.

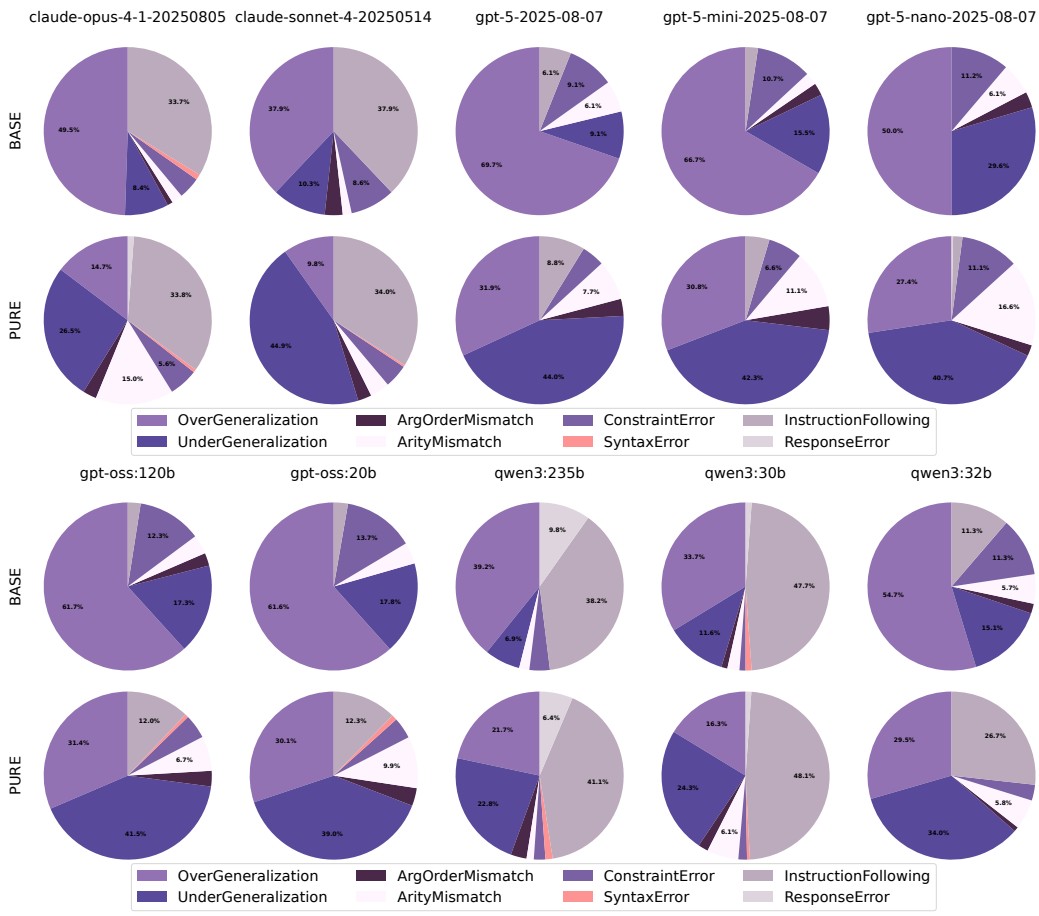

Figure 8: Error analysis of the recent models on TF-Bench.

We then use Table 4 as instructions to prompt GPT-5-mini to analyze the remaining models. In total, we evaluate ten state-of-the-art reasoning models, five proprietary API-access models and five open-access models. The results appear in Figure 8. We observe that models within the same family tend to make similar types of errors. In particular, the Claude, GPT, and Qwen families exhibit notably different error profiles. Both the proprietary GPT models (gpt-5 series) and the open-access GPT models (gpt-oss series) show similar error patterns. The analysis also reveals architectural effects. For example, qwen3:235b and qwen3:30b are MoE models, whereas qwen3:32b is a dense model. Although they share the same training corpus and released at the same time, the MoE models' error distributions differ markedly from the dense model's. We believe this error analysis offers useful insights for future model development, and analysis of reasoning abilities lead by different training strategies and model architectures.

## H  Additional figures

```haskell
words :: String → [String]
words s =  case dropWhile isSpace s of
        "" → []
        s' → w : words s''
          where (w, s'') = break isSpace s'
```

Listing (4) Monomorphic function

```haskell
map :: (a → b) → [a] → [b]
map f [] = []
map f (x:xs) = f x : map f xs
```

Listing (5) Parametric polymorphic

```haskell
(==) :: Eq a ⇒ a → a → Bool
x == y = not (x ⫇ y)
```

Listing (6) Ad-hoc polymorphism

Figure 9: Example function definitions taken from Haskell Standard Prelude [16].

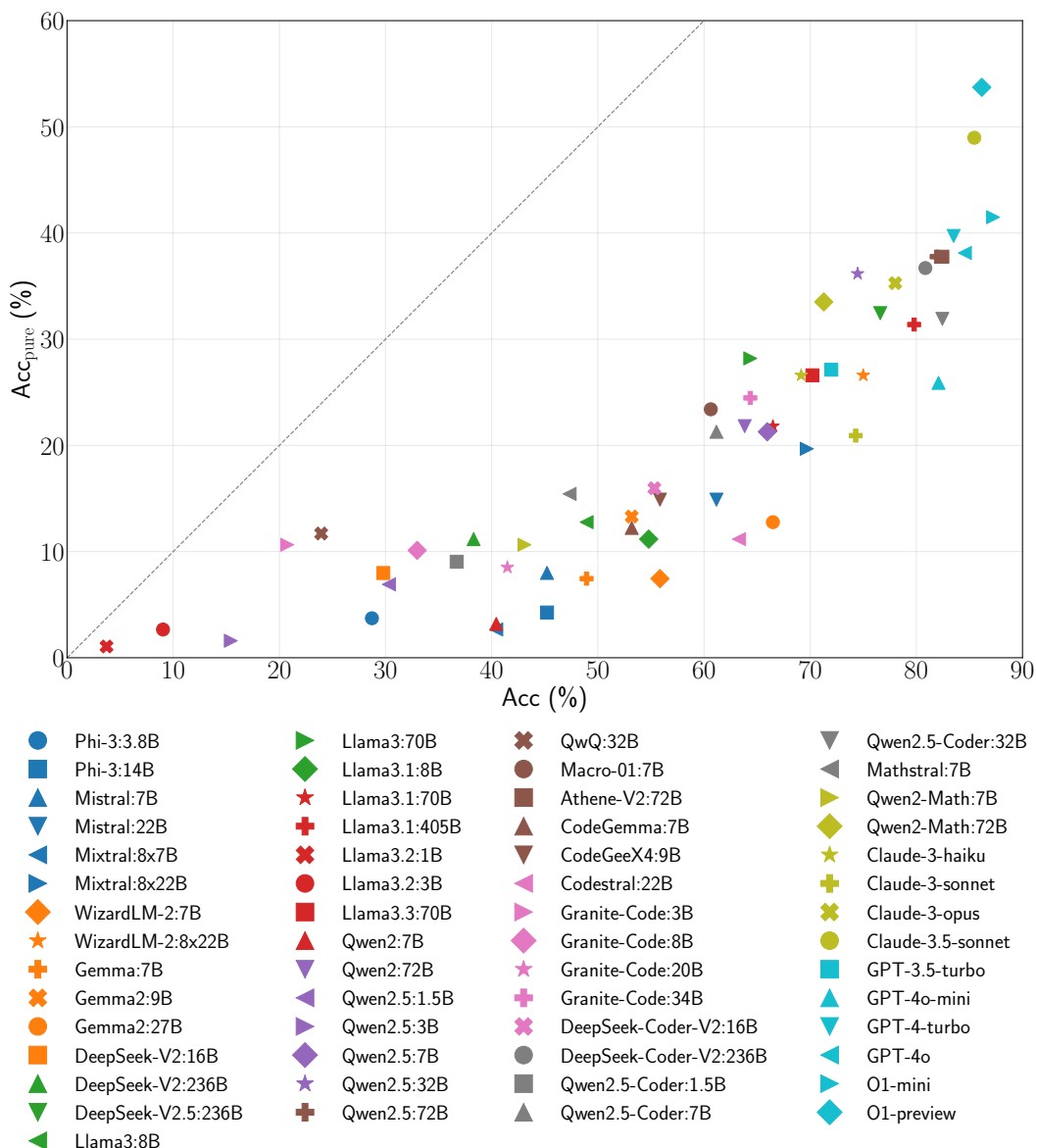

Figure 10: Robustness analysis of models on TF-Bench and TF-Bench$_{pure}$.

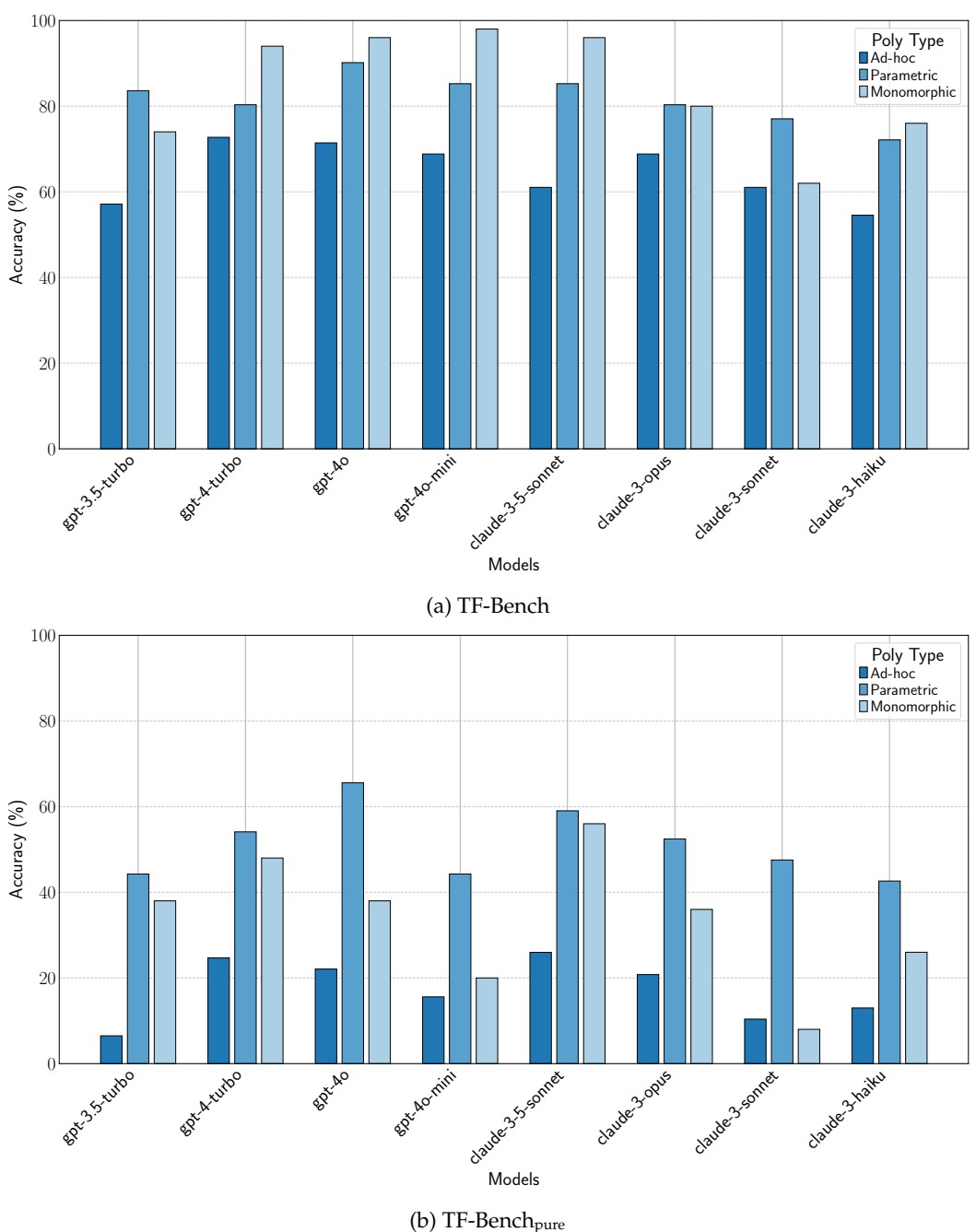

(a) TF-Bench

(b) TF-Bench_pure

Figure 11: Accuracy by category for TF-Bench and TF-Bench_pure.

# I  Additional evaluation results

This section presents supplementary evaluation results for models that are not central to our research questions and were omitted from the main text due to space constraints. We also report results for several models released after submission that may be of interest to readers.

## I.1  API-access models

We also provide additional evaluation results of older API-access models on TF-Bench, as a complementary to Table 1. The results are shown in Table 5.

Table 5: Additional evaluation results of API-access models on TF-Bench. The models are separated by whether they are too old before our paper is submitted for review (top) or newly released after that (bottom).

| Model | Version | TTC | Acc (%) | $Acc_{pure}$(%) |
|---|---|---|---|---|
| Claude-3-haiku | 2024-03-07 | ✗ | 69.15 | 26.60 |
| Claude-3-sonnet | 2024-02-29 | ✗ | 74.29 | 20.92 |
| GPT-3.5-turbo | 0125 | ✗ | 71.99 | 27.13 |
| GPT-4o-mini | 2024-07-18 | ✗ | 82.09 | 25.89 |
| GPT-O1-mini | 2024-09-12 | ✓ | 87.23 | 41.49 |
| GPT-O1-preview | 2024-09-12 | ✓ | 86.17 | 53.72 |
| GPT-5 | 2025-08-07 | ✓ | 83.34 | 51.95 |
| GPT-5-mini | 2025-08-07 | ✓ | 83.34 | 47.52 |
| GPT-5-nano | 2025-08-07 | ✓ | 82.62 | 41.13 |
| Claude-4.1-opus* | 2025-05-14 | ✓ | 83.16 | 39.71 |
| Claude-4-sonnet* | 2025-05-14 | ✓ | 89.72 | 53.01 |

*: We experienced the performance issue of Claude 4 models [80].

One notable observation in Table 5 concerns the three versions of GPT-5. Since these models are proprietary, detailed information about their architectural differences is unavailable. We therefore infer that the mini and nano variants are smaller models distilled from the full GPT-5, likely through strong-to-weak distillation [81] or similar techniques. Interestingly, while the mini and nano models achieve performance comparable to the full GPT-5 on the base split of TF-Bench, they perform substantially worse on TF-Bench$_{pure}$. This pattern suggests that smaller distilled models may capture only superficial knowledge from the teacher model, failing to acquire essential reasoning abilities. Exploring this limitation is a promising direction for future work.

## I.2  Open-access models

We run all the open-access models using Ollama [71]. All the experiments are conducted with a server running Ubuntu 24.04, equipped with 8 NVIDIA H100 GPUs (80GB), and 1.5TB of RAM. The results are shown in Table 6.

Table 6: Full evaluation results of popular open-source LLMs on TF-Bench.
*: The model runs very slowly, we set the timeout of each task to five minutes.

| LLM Type | Model | Size | TTC | Acc (%) | $\text{Acc}_{\text{pure}}$(%) |
|---|---|---|---|---|---|
| OSS | Phi-3 [82] | 3.8B | ✗ | 28.72 | 3.72 |
| | | 14B | ✗ | 45.21 | 4.26 |
| | Mistral [83] | 7B | ✗ | 45.21 | 7.99 |
| | | 22B | ✗ | 61.17 | 14.89 |
| | Mixtral [84] | 8×7B | ✗ | 40.43 | 2.66 |
| | | 8×22B | ✗ | 69.68 | 19.68 |
| | WizardLM-2 [85] | 7B | ✗ | 55.85 | 7.45 |
| | | 8×22B | ✗ | 75.00 | 26.60 |
| | Gemma [3] | 7B | ✗ | 48.94 | 7.45 |
| | Gemma2 [86] | 9B | ✗ | 53.19 | 13.30 |
| | | 27B | ✗ | 66.49 | 12.77 |
| | DeepSeek-V2 [87] | 16B | ✗ | 29.79 | 7.98 |
| | | 236B | ✗ | 38.30 | 11.17 |
| | DeepSeek-V2.5 [87] | 236B | ✗ | 76.60 | 32.45 |
| | Llama3 [88] | 8B | ✗ | 48.94 | 12.77 |
| | | 70B | ✗ | 64.36 | 28.19 |
| | Llama3.1 [89] | 8B | ✗ | 54.79 | 11.17 |
| | | 70B | ✗ | 66.49 | 21.81 |
| | | 405B | ✗ | 79.79 | 31.38 |
| | Llama3.2 [90] | 1B | ✗ | 3.72 | 1.06 |
| | | 3B | ✗ | 9.04 | 2.66 |
| | Llama3.3 [91] | 70B | ✗ | 70.21 | 26.60 |
| | Qwen2 [92] | 7B | ✗ | 40.43 | 3.19 |
| | | 72B | ✗ | 63.83 | 21.81 |
| | Qwen2.5 [92] | 1.5B | ✗ | 30.32 | 6.91 |
| | | 3B | ✗ | 15.43 | 1.60 |
| | | 7B | ✗ | 65.96 | 21.28 |
| | | 32B | ✗ | 74.47 | 36.17 |
| | | 72B | ✗ | 81.91 | **37.77** |
| | QwQ [93] | 32B | ✓ | 23.94* | 11.70* |
| | Marco-o1 [94] | 7B | ✓ | 60.64 | 23.40 |
| | Athene-V2 [95] | 72B | ✗ | **82.45** | **37.77** |
| Code | CodeGemma [96] | 7B | ✗ | 53.19 | 12.23 |
| | CodeGeeX4 [97] | 9B | ✗ | 55.85 | 14.89 |
| | Codestral [98] | 22B | ✗ | 63.30 | 11.17 |
| | Granite-Code [21] | 3B | ✗ | 20.74 | 10.64 |
| | | 8B | ✗ | 32.98 | 10.11 |
| | | 20B | ✗ | 41.48 | 8.51 |
| | | 34B | ✗ | 64.36 | 24.47 |
| | DeepSeek-Coder-V2 [99] | 16B | ✗ | 55.32 | 15.96 |
| | | 236B | ✗ | 80.85 | **36.70** |
| | Qwen2.5-Coder [70] | 1.5B | ✗ | 36.70 | 9.04 |
| | | 7B | ✗ | 61.17 | 21.28 |
| | | 32B | ✗ | **82.45** | 31.91 |
| Math | Mathstral [100] | 7B | ✗ | 47.34 | 15.43 |
| | Qwen2-Math [101] | 7B | ✗ | 43.09 | 10.64 |
| | | 72B | ✗ | **71.28** | **33.51** |

# J    Limitations and Future Work

## J.1    Analysis of fine-tuning

In Section 4.6, our model selection of the base models of Qwen2-Math and Qwen2.5-Coder is not an exact pair, where Qwen2-Math is fine-tuned from Qwen2, while Qwen2.5-Coder is fine-tuned from Qwen2.5. This limitation is due to Qwen2-Coder and Qwen2.5-Math not being available on Ollama. Although the base models are not exactly the same, they share the same architecture. Qwen2.5 was trained on a larger dataset than Qwen2 [70], which does not conflict with our observations from our experiments.

For this analysis, we only focus on Qwen2 and Mistral since they are the only models providing both code and math versions. While our observations may not generalize to *all* models, as we cannot evaluate every model fine-tuned on both code and math datasets, it highlights a notable trend in post-tuning to enhance models' reasoning capabilities.

## J.2    Downstream application

TF-Bench and its variants are designed to evaluate LLMs' foundational reasoning about program semantics. These benchmarks are not intended to determine the applicability of LLMs in specific downstream engineering tasks, but to provide a systematic and principled evaluation methodology of their intelligence level on programming. Our evaluations aim to offer guidance and confidence when choosing to use LLM-powered tools for essential software tasks.

Deductive reasoning has not been well studied in existing literature on LLMs for PLs, yet it is a crucial aspect of intelligence, particularly within formalist philosophy. We believe that evaluating deductive reasoning is essential for understanding the true capabilities of LLMs, especially given the current trend of models utilizing increased test-time compute (TTC). Previous work on evaluation benchmarks has primarily focused on inductive tasks, like few-shot code generation or code completion, aligning with the next-token prediction generation paradigm of LLMs. However, the modern trend toward TTC involves training models to reason step by step in a deductive manner similar to humans, yet no current PL benchmarks adequately evaluate this critical dimension. Consequently, we argue that TF-Bench is necessary to advance LLM development by specifically addressing and assessing their deductive reasoning capabilities.

## J.3    Future work

While our main focus is evaluating the fundamental deductive reasoning capabilities of LLMs, there are significant real-world applications for type inference within TF-Bench. In PLs that feature static and strong typing, traditional rule-based type inference methods face two primary limitations that LLMs can address: undecidability and poor extensibility.

Some type inference and type checking algorithms are undecidable when relying solely on static analysis, particularly in languages that incorporate flexible generics such as structural subtyping or bounded quantification [102]. Existing research has explored various approaches to mitigate these limitations [103]. In these scenarios, LLM-based type inference tools can be particularly beneficial by providing probabilistic type inference results.

Furthermore, applying new rule-based typing tools to existing programming languages remains challenging. Most research in PL tends to create new toy languages to experiment with innovative type system features, rather than extending existing languages. Only a few languages support language-level extensions, and even in those cases, introducing new type system features demands significant effort.

