# OpenReview forum: "Evaluating Program Semantics Reasoning with Type Inference in System $F$"
_NeurIPS.cc/2025/Datasets_and_Benchmarks_Track — NeurIPS 2025 Datasets and Benchmarks Track poster_

### Official Review · Reviewer_6AQC · 2025-07-02

**Rating:** 5
**Confidence:** 4

**Summary:**

The authors present TF-Bench, a benchmark for evaluating LLMs in predicting function types for Haskell. Such a task demands reasoning and program understanding from LLMs. The author's choice of Haskell is motivated by its core design philosophy around System F which allows for easy (and sound) checking of type inference and also since it supports parameter polymorphism which allows for a diverse benchmark.

To makem sure that NL markers are not present in the benchmark dataset, the authors also propose TF-Bench pure which is obtained by pruning NL based syntax in the benchmark dataset.

TF-Bench uses programs from the Haskall Prelude in building the dataset. Tasks are created by processing this codebase and resolving dependencies to provide self-contained setups for each task. For polymorphic functions, generics are retained while also creating new tasks using instantiations.

To minimize (and measure) the impact of NL, the authors introduce 3 rewrite operators (rewriting NL types, variables, binding names) that convert the code into an NL-free program. This would better allow to measure LLM reasoning abilities. The rewritten tasks are validated to ensure that they remain valid and are semantically equivalent.  The benchmarks tests for alpha-equivalence.

The authors then conduct an empirical evaluation on several closed and open LLMs using this benchmark and also fine-tune models to show whether there is a change in performance.

**Dataset Code Accessibility:**

Yes

**Dataset Code Comments:**

The links provided in the appendix seem accessible on Zenodo.

**Ethical Considerations:**

No, there are no or only very minor ethics concerns

**Final Justification:**

Ive read the remainder of the review and the rebuttals. The authors have alleviated my concerns. I think the work is valuable and bridges an important gap in the literature. The paper is clearly written, is well-motivate and has a good empirical design. As a result, I am recommending acceptance.

**Limitations Weaknesses:**

I liked this paper and think it makes good contributions. Im leaning towards an accept but would like some questions answered before I can increase my score to a 5.

1. It is not clear what the dataset size is. Is it only 188 tasks? That seems quite small if it is the case. Please clarify.

2. I think Sec. 4.4 is a bit misleading. You dont really have any adversarial rewrites, so Im not sure why such statements are included in lines 316-318. The definition is fine without mentions of adversarial things and just rewrites are sufficient, although doing this analysis would improve your results.

3. The prompts seem to be zero-shot. Was Chain-of-Thought etc not employed? Also, were few-shot examples provided to the LLM?

4. What is the time complexity of alpha-equivalence? Is it linear or polynomial? Im guessing that its some polynomial since all nested calls must be resolved.

5. Why were other languages like Idris, OCaml etc not considered? These should be languages that LLMs are trained on and are based on similar principles as Haskell. Arguably, Idris has even more functionality.

6. Circling back to #1, given that this is a static benchmark, how will you guard against future generations of LLMs memorizing this new dataset compared to say Jones et al. (as is mentioned in lines 208). Would simple alpha-rewrites safeguard against memorization. This is not well justified I believe.

**Strengths Contributions:**

1. The paper is clear and well-written and easy to understand.

2. The renaming of programs and the choice of Haskell makes sense and I agree that type inference can be construed as evaluating the reasoning abilities of models.

3. The empirical evaluation is solid and the chosen evaluation metrics are decent.

4. The benchmark seems flexible and easy to extend.

---

> ### Author Rebuttal · Authors · 2025-07-27
>
> We thank the reviewer for the constructive comments and questions.
>
> 1. Each variant of TF-Bench contains 188 tasks. These tasks represent the entire Haskell Prelude.
>    We justify only choosing the Prelude in Section 3.1.
>    For summarization, the core library Prelude is extensively reviewed by the language community,
>    which offers quality assurance for the tasks.
>    Furthermore, the semantics of these functions are the most widely used in not only Haskell programs,
>    but also in other functional and imperative languages.
>    Therefore, we consider the Prelude a more representative, reliable, and feasible source for building TF-Bench.
>
> 2. Thank you for pointing out the presentation issue in lines 316–318.
>    We included this sentence because adversarial attacks on code language models also focus on substituting variable names,
>    which is one of our rewriting operators.
>    The only difference is that adversarial attacks optimize the rewritten variable names to maximize the model's prediction error,
>    whereas we remove the NL parts.
>    We will revise this section in the updated version of the paper to focus solely on the rewrites,
>    without referring to adversarial attacks.
>    We plan to explore adversarial analysis in future work.
>
> 3. We evaluate the LLMs in a zero-shot setting because the recent models can already achieve nearly perfect results on the base version of TF-Bench.
>    Therefore, to focus on presenting the true deductive reasoning ability of LLMs by alpha-rewrite,
>    we keep this setting for all experiments.
>    We will clarify this in the final version of the paper.
>
> 4. For alpha-equivalence evaluation, the complexity is actually linear since we only compare the type signatures.
>    We will add this detail in the revised manuscript.
>
> 5. We chose Haskell over other languages with type soundness because its type signatures are decoupled from function implementations,
>    allowing us to construct tasks focused solely on generating the signature without requiring the function body.
>    This feature is not available in similar languages such as OCaml, Scala, Rust, and TypeScript.
>    Additional reasons for this choice are discussed in Section 3.1.
>    The only other suitable language with this syntactic feature is Idris.
>    However, Idris is too niche for evaluating LLMs,
>    as it is not even included in recent popular large-scale pre-training code datasets like The Stack V2 [1].
>    Moreover, since current reasoning LLMs struggle with the three common type inference tasks in TF-Bench,
>    introducing dependent types—on which Idris’s type system is based—would not be beneficial.
>    We will include further discussion of this in the limitations section of the final version of the paper.
>    We will consider Idris tasks for additional difficulty levels in future work.
>
> 6. Alpha-rewrite can be adjusted to support different rewrite paradigms.
>    The current paradigm is described in Section 3.2,
>    where we can also inject random noise into the rewritten parts as a safeguard against memorization.
>    We plan to include this feature in the publicly released version of TF-Bench.
>
> [1] StarCoder 2 and The Stack v2: The Next Generation. ArXiv 2024.

---

### Official Review · Reviewer_x3Eb · 2025-07-02

**Rating:** 6
**Confidence:** 4

**Summary:**

This paper proposes a benchmark for evaluating codeLM's capability on understanding code semantics.
The specific task is to let the model predict types of a given program.
Type information is key to understanding code semantics.
It reflects whether models can understand the foundamental logics of a program.

More importantly, evaluating model's capability on type-inference decouples the effects of natural language since type-system is a formal system closer to math deduction rules. It provides valuable insights on how well models can reason about code without relying on natural language hints and thus are complementary to existing code reasoning benchmarks.

I believe the dataset will provide meaningful insights to the community of code reasoning models.

**Additional Feedback:**

It would be better to measure the diversity of data samples. (Such as distributions of program structures, distribution of complexity, etc.)

**Dataset Code Accessibility:**

Yes

**Dataset Code Comments:**

The authors provide both github links and huggingface dataset.

The code quality is good. It provides code for both data augmentation and evaluation.

The dataset is clear and easy to use.

**Ethical Considerations:**

No, there are no or only very minor ethics concerns

**Final Justification:**

I read. the authors response. They clarified the minor questions in my review, and I believe the relevant edits can easily be applied during the camera-ready process. Their response do not induce further concerns.

Therefore, I keep my original score.

**Limitations Weaknesses:**

The model may not be sufficiently trained on Haskell. Why not using more frequent languages such as Python? (Python also has higher-order functions, etc. that can introduce non-trivial typing tasks.)

It would be better to provide more insights on the challenging parts that the model performs poorly, such as code complexity, higher-order functions, etc.

**Strengths Contributions:**

It is a very meaningful benchmark that provide new perspectives on evaluating codeLM's capability.
It decouples the effects of natural language, thus providing insights on model's fundamental reasoning capabilities.

The implementation is neat and solid. It bases on the foundation of program semantics and uses formal approaches to ensure the quality of the benchmark.
The implementation introduces different variants to decouple the effects of NL artifacts, such as variable names.

---

> ### Author Rebuttal · Authors · 2025-07-27
>
> We wish to thank the reviewer for their time and careful reading of our manuscript.
>
> 1. The reason for not choosing other more common languages is due to considerations of type soundness.
>    Languages like C/C++ and Python have either weak or dynamic type systems,
>    meaning that their documented types do not always match the actual types of the functions in terms of semantics.
>    We discuss the reason for choosing Haskell in Section 3.1,
>    and we will add further discussion in the limitations section of the final version of the paper.
>
> 2. Thanks for your suggestion on analyzing challenging parts of the tasks.
>    We included the discussion of performance across the three classes in the appendix (Figure 8)
>    due to space constraints.
>    Figure 8 demonstrates the difficulty of the three classes,
>    where Ad-hoc polymorphism (bounded quantification) is the most difficult.
>    We will revise the manuscript to clearly discuss this in the main text in the final version.

---

### Official Review · Reviewer_96xZ · 2025-07-03

**Rating:** 5
**Confidence:** 2

**Summary:**

This paper introduces TF-Bench, a benchmark evaluating large language models' (LLMs) deductive reasoning for program semantics via type inference in System F, using Haskell tasks; it includes TF-Bench_pure to strip natural language cues, revealing models like Claude-3.7-sonnet achieve only 55.85% accuracy on pure semantics, with novel metrics (semantic robustness and reasoning effectiveness) quantifying robustness and test-time compute improvements, while open-source data on Hugging Face/Zenodo ensures reproducibility, highlighting critical gaps in LLMs' reasoning capabilities.

**Dataset Code Accessibility:**

Partly

**Ethical Considerations:**

No, there are no or only very minor ethics concerns

**Final Justification:**

After reviewing the authors' rebuttal, I am raising my score from 4 to 5 (Accept). The authors have satisfactorily addressed all major concerns: they provided compelling justification for choosing Haskell (unique syntax feature unavailable in other typed languages), committed to including human performance analysis and error categorization, clarified that cross-category analysis exists in the appendix, and substantiated the Prelude dating claim with concrete references (1998 to 2010). The paper's core contributions remain strong, as the formal grounding in System F, novel metrics (RS and RE), and comprehensive evaluation across 64 models represent significant advances in evaluating LLM reasoning capabilities. The benchmark addresses a critical gap in understanding whether LLMs truly reason about program semantics versus exploiting surface patterns, making it a valuable contribution to the community.

**Limitations Weaknesses:**

1. Metric Limitations​​.


**​​RS Ignores Failure Modes​​**: Semantic Robustness (RS) measures performance drop but does not diagnose why models fail (e.g., syntax confusion vs. logic errors). Table 1 reports RS values without error analysis (§4.4). Suggestion: Add qualitative analysis of failure cases (e.g., confusion with t1 vs. T1).


**​​RE Dependency on TTC Toggle​​**: Reasoning Effectiveness (RE) requires manual TTC control (§4.5), excluding models without this feature (e.g., most open-source LLMs). Suggestion: Propose RE alternatives for non-API models.

​2. Untested Human Interpretability​​: While alpha-rewrites preserve compiler-verifiable semantics (§3.2), the paper does not verify if LLMs interpret rewritten code equivalently to original code. Renaming map to f1 may disproportionately disrupt model attention mechanisms. Suggestion: Conduct human/user studies to validate rewrite interpretability.


3. Prelude-Centric Tasks​​: 188 tasks are sourced from Haskell Prelude (§3.1), which may overrepresent common functional patterns (e.g., monads) while underrepresenting complex real-world scenarios (e.g., concurrent or I/O-heavy logic). Suggestion: Incorporate tasks from Hackage packages to diversify task complexity.

**Strengths Contributions:**

1. A major contribution is the proposal of two principled evaluation metrics. Semantic robustness (RS) quantifies model vulnerability to NL perturbations, exposing how heavily current LLMs rely on superficial patterns. Reasoning effectiveness (RE) provides unprecedented insight into test-time compute methods, measuring whether their improvements stem from true reasoning gains or benchmark contamination. The stark performance drop on TF-Bench_pure reveals a fundamental gap even in state-of-the-art models.

2. The paper's empirical depth is exceptional. Comprehensive evaluation across 64 LLMs demonstrates Claude-3.7-sonnet achieves merely 55.85% accuracy on pure semantics tasks. The fine-tuning analysis yields actionable insights, showing math-tuned models significantly outperform code-tuned counterparts in reasoning capabilities—a finding with practical implications for LLM development.

3. Methodologically, the work clearly differentiates itself from prior benchmarks. Unlike Python type annotation tools, it builds on System F's sound formal foundations. Against mathematical reasoning benchmarks, it offers verifiable semantics-preserving perturbations rather than heuristic substitutions. The implementation exemplifies reproducibility through open-sourced datasets, Croissant metadata, and detailed transformation pipelines.

---

> ### Author Rebuttal · Authors · 2025-07-27
>
> We wish to thank the reviewer for the helpful suggestions.
>
> 1. We appreciate the suggestion regarding common error analysis.
>    In the final version of our manuscript,
>    we will include an analysis of the API models for which the reasoning chains are available,
>    allowing us to inspect their reasoning processes.
>    We will also categorize the common error patterns observed across the models.
>    About RE for open-source LLMs without manual TTC control,
>    we will consider adding other inference frameworks (for example, vLLM) to enable this feature.
>
> 2. Thank you for pointing out the human performance analysis.
>    We agree that it could provide a clearer understanding of the task difficulty.
>    If applicable, we will include a small-scale student experiment in the final version of the paper.
>
> 3. Our focus in this benchmark is to highlight a novel aspect of TTC LLMs' deductive reasoning ability,
>    where scenarios like IO operations do not tightly reflect the true reasoning capabilities of LLMs.
>    While TF-Bench is not directly intended for downstream applications,
>    it provides insight into the capacity of LLMs to perform true reasoning.
>    Although our primary goal is to evaluate the fundamental deductive reasoning capabilities of LLMs,
>    type inference within TF-Bench also has significant real-world applications.
>    We include this discussion in Appendix B.2.

---

> > ### Comment · Reviewer_96xZ · 2025-08-05
> >
> > Thanks for the clarifications. I appreciate the planned error analysis, vLLM consideration for open-source models, and human performance study. I now understand the focus on fundamental deductive reasoning rather than downstream applications. Two remaining questions:
> >
> > (1) Will the error categorization distinguish between syntax confusion (e.g., t1 vs T1) versus logical reasoning failures?
> >
> > (2) Any plans to expand beyond Prelude tasks in future work?
> >
> > The response addresses most of my concerns and I look forward to the enhanced analysis in the final version.

---

> > > ### Author Response · Authors · 2025-08-05
> > >
> > > Thank you for your reply and further discussion of the paper.
> > >
> > > 1. Yes, we will distinguish between syntax confusion and logical reasoning failures.
> > >    Furthermore, we will analyze whether cases such as t1 v.s. T1 are due to syntactic confusion
> > >    or whether the LLM fails to reason about polymorphism
> > >    (since t1 is a generic type variable and T1 is a concrete type).
> > >    Logical reasoning failures apply only to models that use explicit reasoning chains,
> > >    which we will do our best to analyze in the final version of the paper.
> > >
> > > 2. Yes, we will extend our work to include other languages and packages in the future.
> > >    Currently, dependency solving is performed using Hoogle's API,
> > >    which only supports functions from the Prelude.
> > >    In future work, we plan to extend dependency solving to other packages using the build systems and static analysis tools.
> > >    To be more specific, this will involve downloading all required packages locally using the Stack build system,
> > >    then using the Language Server Protocol to resolve dependencies in type signatures, as in previous work [1].
> > >
> > > [1]: He et al. "UniTSyn: A Large-Scale Dataset Capable of Enhancing the Prowess of Large Language Models for Program Testing." ISSTA 2024.

---

> > > > ### Comment · Reviewer_96xZ · 2025-08-08
> > > >
> > > > Thank you for the detailed response. I have no further concerns and appreciate your thorough engagement with the review feedback.

---

### Official Review · Reviewer_Dy6S · 2025-07-07

**Rating:** 4
**Confidence:** 4

**Summary:**

The paper introduces TF-Bench, a benchmark aimed at evaluating program semantics reasoning in LLMs via type inference tasks in System F, using Haskell as the underlying language. It proposes two variants: (i) TF-Bench, with NL elements like variable names and types, and (ii) TF-Benchpure, where these elements are abstracted away using verified rewrites. The benchmark evaluates LLMs' reasoning capabilities isolated from surface-level language cues. The authors evaluate 64 LLMs on this task, propose two new metrics, semantic robustness and reasoning effectiveness, and analyze the influence of training data (code vs. math) on reasoning performance.

**Dataset Code Accessibility:**

Yes

**Ethical Considerations:**

No, there are no or only very minor ethics concerns

**Limitations Weaknesses:**

- The benchmark is restricted to Haskell only, and although Haskell’s type system is well-suited for such reasoning tasks, the paper does not engage with whether the insights generalize to other typed languages (e.g., OCaml, Scala, Rust, TypeScript). I feel that the authors should either broaden the scope or critically discuss this limitation.

- A small-scale pilot human performance analysis with students or developers would help calibrate task difficulty, have an interpretive anchor and clarify whether LLM limitations reflect a real challenge or just poor adaptation.

- The paper lacks error analysis or categorization of reasoning failures (e.g., incorrect inference chains, misapplication of ad-hoc polymorphism, etc.). This could provide deeper insights and inform model or benchmark design.

- While the tasks include three classes (monomorphic, parametric, ad-hoc), there is no discussion of performance across these dimensions. How much harder is ad-hoc polymorphism? How does RE or RS vary by function type? Without this, conclusions about "reasoning" remain too coarse.

- The authors mention the Haskell Prelude predates model cutoffs, but do not provide empirical evidence (e.g., retrieval analysis, nearest-neighbor overlap). This is particularly important given models may memorize standard library code.

- The paper makes strong claims about reasoning and robustness, but does not clearly connect these findings to real-world developer use cases or tooling scenarios. I am curious if this benchmark predict LLM performance in static analysis, debugging, or formal verification?

- Some tables (e.g., Table 3) are dense and could benefit from clearer structuring. Figures are useful but could be expanded to visualize score distributions or per-category accuracy.

**Strengths Contributions:**

- The paper focuses on a formally grounded and language-centric deductive task, type inference in System F, as a proxy for evaluating program semantics reasoning in LLMs. This is well-justified as a minimal, verification-friendly task that aligns with the Curry-Howard correspondence (Section 2.2).

- The creation of TF-Benchpure is a meaningful step forward: by stripping NL elements (names, types, bindings), the authors effectively neutralize spurious correlations (Section 3.2). This isolates the semantic core of the reasoning task.

- The authors ensure task soundness by explicitly resolving type dependencies (Section 3.1), verifying compilability, and validating tasks via Hindley-Milner inference and manual inspection. This rigor supports reliable evaluation.

- Semantic Robustness (RS) and Reasoning Effectiveness (RE) are principled, well-motivated additions that provide interpretable and meaningful insights. In particular, RE provides a fresh angle to disentangle memorization from actual generalization in test-time compute (TTC) models.

- The benchmark is applied across 64 LLMs spanning commercial APIs and OSS models, making the analysis timely and broadly informative. Evaluation is meticulous (multiple seeds, standard errors, prompt normalizations). The paper shows that fine-tuning on math datasets yields better reasoning gains than on code datasets, an important and underappreciated point. The experimental evidence supports this claim clearly.

---

> ### Author Rebuttal · Authors · 2025-07-27
>
> We thank the reviewer for taking the time to carefully read our manuscript.
>
> 1. We chose Haskell over other languages with type soundness because
>    its type signatures are decoupled from function implementations,
>    allowing us to construct tasks focused solely on generating the signature without requiring the function body.
>    This feature is not available in similar languages such as OCaml, Scala, Rust, and TypeScript.
>    Additional reasons for this choice are discussed in Section 3.1.
>    The only other suitable language with this syntax feature is Idris.
>    However, Idris is too niche for evaluating LLMs,
>    which is not even included in recent popular large-scale pre-training code datasets like The Stack V2 [1].
>    Moreover, since current reasoning LLMs struggle with the three common type inference tasks in TF-Bench,
>    introducing dependent types, on which Idris's type system is based, would not be beneficial.
>    We will include further discussion of this in the limitations section of the final version of the paper.
>
> 2. Thank you for pointing out the human performance analysis.
>    We agree that it could provide a clearer understanding of the task difficulty.
>    If applicable, we will include a small-scale student experiment in the final version of the paper.
>
> 3. We appreciate the suggestion regarding common error analysis.
>    In the final version of our manuscript,
>    we will include an analysis of the API models for which the reasoning chains are available,
>    allowing us to inspect their reasoning processes.
>    We will also categorize the common error patterns observed across the models.
>
> 4. We included the discussion of performance across the three classes in the appendix (Figure 8)
>    due to space constraints.
>    In the final version of the paper, we will expand this analysis to include results on RE and SR as well.
>
> 5. The current Standard Prelude was introduced in 1998 [2],
>    with the current version published in 2010 [3].
>    The source code of the Prelude has been publicly available online on GitLab and GitHub since then.
>    Based on this evidence, we note that the Haskell Prelude predates the LLMs evaluated in this paper.
>    We will add these references as supporting evidence in the final version of the paper.
>
> 6. Our focus in this benchmark is to highlight a novel aspect of TTC LLMs' deductive reasoning ability.
>    While TF-Bench is not directly intended for downstream applications,
>    it provides insight into the capacity of LLMs to perform true reasoning.
>    Although our primary goal is to evaluate the fundamental deductive reasoning capabilities of LLMs,
>    type inference within TF-Bench also has significant real-world applications.
>    We include this discussion in Appendix B.2.
>
> 7. Thank you for pointing out the visuals of our tables and figures.
>    We will consider splitting the tables to improve clarity, if space permits in the final version.
>    Per-category accuracy and score distribution are provided in Appendix Figures 7 and 8,
>    and we will ensure these are clearly referenced in the main text in the revised version.
>
> [1] StarCoder 2 and The Stack v2: The Next Generation. ArXiv 2024.
>
> [2] Simon Peyton Jones. Haskell 98 language and libraries: the revised report. Cambridge University Press, 2003.
>
> [3] Simon Marlow, et. al. Haskell 2010 Language Report. Online at http://www.haskell.org

---

### Decision · Program_Chairs · 2025-09-18

**Decision:**

Accept (poster)

**Comment:**

This paper introduces TF-Bench, a benchmark designed to evaluate the deductive reasoning capabilities of Large Language Models (LLMs) on program semantics. The core task is type inference in System F. The authors also introduced TF-Bench_pure, a variant where semantically irrelevant natural language elements are systematically removed through verified transformations. This allows for an evaluation of a model's pure semantic reasoning, disentangled from superficial token associations. The findings show that LLMs such as Claude-3.7-sonnet would struggle with the pure semantics version. The authors also propose two new metrics, Semantic Robustness (RS) and Reasoning Effectiveness (RE), to measure a model's resilience to semantic-preserving perturbations and the true benefit of test-time compute, respectively.

Reviewers generally like the comprehensive analysis and the solid benchmark created. During rebuttal the authors have clarified several concerns around the particular choice of Haskell, and I strongly encourage the authors to include additional results like human performance study as mentioned during the rebuttal.